# Object Localization based on Structural SVM using Privileged Information

**Jan Feyereisl, Suha Kwak**[*]**, Jeany Son, Bohyung Han**
Dept. of Computer Science and Engineering, POSTECH, Pohang, Korea
`thefillm@gmail.com, {mercury3,jeany,bhhan}@postech.ac.kr`

## Abstract

We propose a structured prediction algorithm for object localization based on Support Vector Machines (SVMs) using privileged information. Privileged information provides useful high-level knowledge for image understanding and facilitates learning a reliable model even with a small number of training examples. In our setting, we assume that such information is available only at training time since it may be difficult to obtain from visual data accurately without human supervision. Our goal is to improve performance by incorporating privileged information into ordinary learning framework and adjusting model parameters for better generalization. We tackle object localization problem based on a novel structural SVM using privileged information, where an alternating loss-augmented inference procedure is employed to handle the term in the objective function corresponding to privileged information. We apply the proposed algorithm to the Caltech-UCSD Birds 200-2011 dataset, and obtain encouraging results suggesting further investigation into the benefit of privileged information in structured prediction.

## 1 Introduction

Object localization is often formulated as a binary classification problem, where a learned classifier determines the existence or absence of a target object within a candidate window of every location, size, and aspect ratio. Recently, a structured prediction technique using Support Vector Machine (SVM) has been applied to this problem [1], where the optimal bounding box containing target object is obtained by a trained classifier. This approach provides a unified framework for detection and post-processing (non-maximum suppression), and handles issues related to the object with variable aspect ratios naturally. However, object localization is an inherently difficult task due to the large amount of variations in objects and scenes, *e.g.*, shape deformations, color variations, pose changes, occlusion, view point changes, background clutter, etc. This issue is aggravated when the size of training dataset is small.

More reliable model can be learned even with fewer training examples if additional high-level knowledge about an object of interest is available during training. Such high-level knowledge is called privileged information, which typically describes useful semantic properties of an object such as parts, attributes, and segmentations. This idea corresponds to the Learning Using Privileged Information (LUPI) paradigm [3], which exploits the additional information to improve predictive models in training but does not require the information for prediction. The LUPI framework has been incorporated into SVM in the form of the SVM+ algorithm [4]. However, the applications of SVM+ are often limited to binary classification problems [3, 4].

We propose a novel Structural SVM using privileged information (SSVM+) framework, shown in Figure 1, and apply the algorithm to the problem of object localization. In this formulation, privileged information, *e.g.*, parts, attributes and segmentations, are incorporated to learn a structured

---

[*]Current affiliation: INRIA–WILLOW Project, Paris, France; e-mail: `suha.kwak@inria.fr`

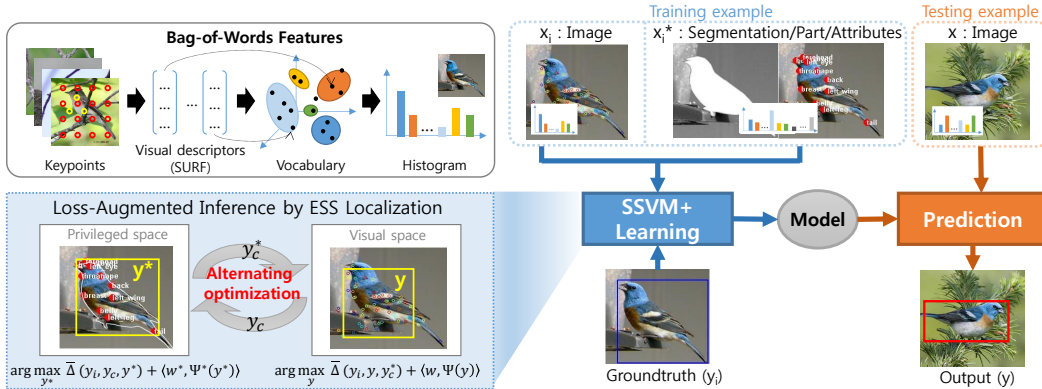

Figure 1: Overview of our object localization framework using privileged information. Unlike visual observations, privileged information is available only during training. We use attributes and segmentation masks of an object as privileged information to improve generalization of trained model. To incorporate privileged information during training, we propose an extension of SSVM, called SSVM+, whose loss-augmented inference is performed by alternating Efficient Subwindow Search (ESS) [2].

prediction function for object localization. Note that high-level information is available only for training but not testing in this framework. Our algorithm employs an efficient branch-and-bound loss-augmented subwindow search procedure to perform the inference by a joint optimization in original and privileged spaces during training. Since the additional information is not used in testing, the inference in testing phase is the same as the standard Structural SVM (SSVM) case. We evaluate our method by learning to localize birds in the Caltech-UCSD Birds 200-2011 (CUB-2011) dataset [5] and exploiting attributes and segmentation masks as privileged information in addition to standard visual features. The main contributions of our work are as follows:

- We introduce a novel framework for object localization exploiting privileged information that is not required or needed to be inferred at test time.

- We formulate an SSVM+ framework, where an alternating loss-augmented inference procedure for efficient subwindow search is incorporated to handle the privileged information together with the conventional visual features.

- Performance gains in localization and classification are achieved, especially with small training datasets.

Methods that exploit additional information have been discussed to improve models for image classification or search in the context of transfer learning [6, 7], learning with side information [8, 9, 10] and domain adaptation [11], where underlying techniques rely on pair-wise constraints [8], multiple kernels [9] or metric learning [9]. Zero-shot learning is an extreme framework, where the models for unseen classes are constructed even without training data [12, 13]. Recent works often rely on natural language processing techniques to handle pure textual description [14, 15].

Standard learning algorithms require many data to construct a robust model while zero-shot learning does not need any training examples. LUPI framework is in the middle of traditional data-driven learning and zero-shot learning since it aims to learn a good model with a small number of training data by taking advantage of privileged information available at training time. Privileged information has been considered in face recognition [16], facial feature detection [17], and event recognition [18], but such works are still uncommon. Our work applies the LUPI framework to an object localization problem based on SSVM. The use of SSVMs for object localization is originally investigated by [1]. More recently, [19, 20] employ SSVM as part of their localization procedure, however none of them incorporate privileged information or similar idea. Recently, [21] presented the potential benefit of SVM+ in object recognition task.

The rest of this paper is organized as follows. We first review the LUPI framework and SSVM in Section 2, and our SSVM+ formulation for object localization is presented in Section 3. The performance of our object localization algorithm is evaluated in Section 4.

## 2   Background

### 2.1   Learning Using Privileged Information

The LUPI paradigm [3, 4, 22, 23] is a framework for incorporating additional information during training that is not available at test time. The inclusion of such information is exploited to find a better model, which yields lower generalization error. Contrary to classical supervised learning, where pairs of data are provided $(\boldsymbol{x}_1, y_1), \ldots, (\boldsymbol{x}_n, y_n)$, $\boldsymbol{x}_i \in \mathcal{X}$, $y_i \in \{-1, 1\}$, in the LUPI paradigm additional information $\boldsymbol{x}^* \in \mathcal{X}^*$ is provided with each training example as well, *i.e.*, $(\boldsymbol{x}_1, \boldsymbol{x}_1^*, y_1), \ldots, (\boldsymbol{x}_n, \boldsymbol{x}_n^*, y_n)$, $\boldsymbol{x}_i \in \mathcal{X}$, $\boldsymbol{x}_i^* \in \mathcal{X}^*$, $y_i \in \{-1, 1\}$. This information is, however, not required during testing. In both learning paradigms, the task is then to find among a collection of functions the one that best approximates the underlying decision function from the given data.

Specifically, we formulate object localization within a LUPI framework as learning a pair of functions $h : \mathcal{X} \mapsto \mathcal{Y}$ and $\phi : \mathcal{X}^* \mapsto \mathcal{Y}$ jointly, where only $h$ is used for prediction. These functions, for example, map the space of images and attributes to the space of bounding box coordinates $\mathcal{Y}$. The decision function $h$ and the correcting function $\phi$ depend on each other by the following relation,

$$\forall\, 1 \le i \le n, \quad \ell_{\mathcal{X}}(h(\boldsymbol{x}_i), y_i) \le \ell_{\mathcal{X}^*}(\phi(\boldsymbol{x}_i^*), y_i), \tag{1}$$

where $\ell_{\mathcal{X}}$ and $\ell_{\mathcal{X}^*}$ denote the empirical loss functions on the visual ($\mathcal{X}$) and the privileged space ($\mathcal{X}^*$), respectively. This inequality is inspired by the LUPI paradigm [3, 4, 22, 23], where for all training examples the model $h$ is always corrected to have a smaller loss on data than the model $\phi$ on privileged information. The constraint in Eq. (1) is meaningful when we assume that, for the same number of training examples, the combination of visual and privileged information provides a space to learn a better model than visual information alone.

To translate this general learning idea into practice, the SVM+ algorithm for binary classification has been developed [3, 4, 22]. The SVM+ algorithm replaces the slack variable $\xi$ in the standard SVM formulation by a correcting function $\xi = (\langle \boldsymbol{w}^*, \boldsymbol{x}^* \rangle + d)$, which estimates its values from the privileged information. This results in the following formulation,

$$\min_{\boldsymbol{w}, \boldsymbol{w}^*, b, b^*}\; \frac{1}{2}\|\boldsymbol{w}\|_2^2 + \frac{\gamma}{2}\|\boldsymbol{w}^*\|_2^2 + \frac{C}{n} \sum_{i=1}^{n} \underbrace{(\langle \boldsymbol{w}^*, \boldsymbol{x}_i^* \rangle + b^*)}_{\xi_i}, \tag{2}$$

$$\text{s.t.}\quad y_i(\langle \boldsymbol{w}, \boldsymbol{x}_i \rangle + b) \ge 1 - \underbrace{(\langle \boldsymbol{w}^*, \boldsymbol{x}_i^* \rangle + b^*)}_{\xi_i}, \quad \underbrace{(\langle \boldsymbol{w}^*, \boldsymbol{x}_i^* \rangle + b^*)}_{\xi_i} \ge 0, \quad \forall\, 1 \le i \le n,$$

where the terms $\boldsymbol{w}^*, \boldsymbol{x}^*$ and $b^*$ play the same role as $\boldsymbol{w}, \boldsymbol{x}$ and $b$ in the classical SVM, however within the new correcting space $\mathcal{X}^*$. Furthermore, $\gamma$ denotes a regularization parameter for $\boldsymbol{w}^*$. It is important to observe that the weight vector $\boldsymbol{w}$ depends not only on $\boldsymbol{x}$ but also on $\boldsymbol{x}^*$. For this reason the function that replaces the slack $\xi$ is called the *correcting* function. As privileged information is only used to estimate the values of the slacks, it is required only during training but not during testing. Theoretical analysis [4] shows that the bound on the convergence rate of the above SVM+ algorithm could substantially improve upon standard SVM if suitable privileged information is used.

### 2.2   Structural SVM (SSVM)

SSVMs discriminatively learn a weight vector $\boldsymbol{w}$ for a scoring function $f : \mathcal{X} \times \mathcal{Y} \mapsto \mathbb{R}$ over the set of training input/output pairs. Once learned, the prediction function $h$ is obtained by maximizing $f$ over all possible $\boldsymbol{y} \in \mathcal{Y}$ as follows:

$$\hat{\boldsymbol{y}} = h(\boldsymbol{x}) = \arg\max_{\boldsymbol{y} \in \mathcal{Y}} f(\boldsymbol{x}, \boldsymbol{y}) = \arg\max_{\boldsymbol{y} \in \mathcal{Y}} \langle \boldsymbol{w}, \Psi(\boldsymbol{x}, \boldsymbol{y}) \rangle, \tag{3}$$

where $\Psi : \mathcal{X} \times \mathcal{Y} \to \mathbb{R}^d$ is the joint feature map that models the relationship between input $\boldsymbol{x}$ and structured output $\boldsymbol{y}$. To learn the weight vector $\boldsymbol{w}$, the following optimization problem (margin-rescaling) then needs to be solved:

$$\min_{\boldsymbol{w}, \xi}\; \frac{1}{2}\|\boldsymbol{w}\|^2 + \frac{C}{n} \sum_{i=1}^{n} \xi_i, \tag{4}$$

$$\text{s.t.}\quad \langle \boldsymbol{w}, \delta\Psi_i(\boldsymbol{y}) \rangle \ge \Delta(\boldsymbol{y}_i, \boldsymbol{y}) - \xi_i \quad 1 \le i \le n,\; \forall \boldsymbol{y} \in \mathcal{Y},$$

where $\delta\Psi_i(\boldsymbol{y}) \equiv \Psi(\boldsymbol{x}_i, \boldsymbol{y}_i) - \Psi(\boldsymbol{x}_i, \boldsymbol{y})$, and $\Delta(\boldsymbol{y}_i, \boldsymbol{y})$ is a task-specific loss that measures the quality of the prediction $\boldsymbol{y}$ with respect to the ground-truth $\boldsymbol{y}_i$. To obtain a prediction, we need to maximize Eq. (3) over the response variable $\boldsymbol{y}$ for a given input $\boldsymbol{x}$. SSVMs are a general method for solving a variety of prediction tasks. For each application, the joint feature map $\Psi$, the loss function $\Delta$ and an efficient loss-augmented inference technique need to be customized.

## 3   Object Localization with Privileged Information

We deal with object localization with privileged information: given a set of training images of objects, their locations and their attribute and segmentation information, we want to learn a function to localize objects of interest in yet unseen images. Unlike existing methods, our learned function does not need explicit or even inferred attribute and segmentation information during prediction.

### 3.1   Structural SVM with Privileged Information (SSVM+)

We extend the above structured prediction problem to exploit privileged information. Recollecting Eq. (1), to learn the pair of interdependent functions $h$ and $\phi$, we learn to predict a structure $\boldsymbol{y}$ based on a training set of triplets, $(\boldsymbol{x}_1, \boldsymbol{x}_1^*, \boldsymbol{y}_1), \ldots, (\boldsymbol{x}_n, \boldsymbol{x}_n^*, \boldsymbol{y}_n)$, $\boldsymbol{x}_i \in \mathcal{X}, \boldsymbol{x}_i^* \in \mathcal{X}^*, \boldsymbol{y}_i \in \mathcal{Y}$, where $\mathcal{X}$ corresponds to various visual features, $\mathcal{X}^*$ to attributes or segmentations, and $\mathcal{Y}$ is the space of all possible bounding boxes. Once learned, only the function $h$ is used for prediction. It is obtained by maximizing the learned function over all possible joint features based on input $\boldsymbol{x} \in \mathcal{X}$ and output $\boldsymbol{y} \in \mathcal{Y}$ as in Eq. (3), identically to standard SSVMs.

On the other hand, to jointly learn $h$ and $\phi$, subject to the constraint in Eq. (1), we need to extend the SSVM framework substantially. The functions $h$ and $\phi$ are characterized by the parameter vectors $\boldsymbol{w}$ and $\boldsymbol{w}^*$, respectively as

$$h(\boldsymbol{x}) = \arg\max_{\boldsymbol{y}\in\mathcal{Y}}\langle\boldsymbol{w}, \Psi(\boldsymbol{x},\boldsymbol{y})\rangle \quad \text{and} \quad \phi(\boldsymbol{x}^*) = \arg\max_{\boldsymbol{y}^*\in\mathcal{Y}}\langle\boldsymbol{w}^*, \Psi(\boldsymbol{x}^*,\boldsymbol{y}^*)\rangle. \tag{5}$$

To learn the weight vectors $\boldsymbol{w}$ and $\boldsymbol{w}^*$ simultaneously, we propose a novel max-margin structured prediction framework called SSVM+ that incorporates the constraint in Eq. (1) and hence learns two models jointly as follows:

$$\min_{\boldsymbol{w},\boldsymbol{w}^*,\xi} \ \frac{1}{2}\|\boldsymbol{w}\|^2 + \frac{\gamma}{2}\|\boldsymbol{w}^*\|^2 + \frac{C}{n}\sum_{i=1}^{n}\xi_i, \tag{6}$$

$$\text{s.t.} \quad \langle\boldsymbol{w}, \delta\Psi_i(\boldsymbol{y})\rangle + \langle\boldsymbol{w}^*, \delta\Psi_i^*(\boldsymbol{y}^*)\rangle \geq \bar{\Delta}(\boldsymbol{y}_i, \boldsymbol{y}, \boldsymbol{y}^*) - \xi_i \quad \forall 1 \leq i \leq n, \ \forall \boldsymbol{y}, \boldsymbol{y}^* \in \mathcal{Y}.$$

where $\delta\Psi_i^*(\boldsymbol{y}^*) \equiv \Psi^*(\boldsymbol{x}_i^*, \boldsymbol{y}_i) - \Psi^*(\boldsymbol{x}_i^*, \boldsymbol{y}^*)$ and the inequality in Eq. (1) is introduced via a surrogate task-specific loss $\bar{\Delta}$ derived from [23]. This surrogate loss is defined as

$$\bar{\Delta}(\boldsymbol{y}_i, \boldsymbol{y}, \boldsymbol{y}^*) = \frac{1}{\rho}\Delta^*(\boldsymbol{y}_i, \boldsymbol{y}^*) + [\Delta(\boldsymbol{y}_i, \boldsymbol{y}) - \Delta^*(\boldsymbol{y}_i, \boldsymbol{y}^*)]_+, \tag{7}$$

where $[t]_+ = \max(t, 0)$ and $\rho > 0$ is a penalization parameter corresponding to the constraint in Eq. (1), and task-specific loss functions $\Delta$ and $\Delta^*$ are defined in Section 3.3. Through this surrogate loss, we can apply the inequality in Eq. (1) within the ordinary max-margin optimization framework.

Our framework enforces that the model learned on attributes and segmentations ($\boldsymbol{w}^*$) always corrects the model trained on visual features ($\boldsymbol{w}$). This results in a model with better generalization on visual features alone. Similar to SSVMs, we can tractably deal with the exponential number of possible constraints present in our problem via loss-augmented inference and optimization methods such as the cutting plane algorithm [24] or the more recent block-coordinate Frank Wolfe method [25]. Pseudocode for solving Eq. (6) using the the cutting plane method is presented in Algorithm 1.

Our formulation has a general form that follows the SSVM framework. This means that Eq. (6) is independent of the definitions of joint feature map, task-specific loss and loss-augmented inference. We can therefore apply our method to a variety of other problems in addition to object localization. All that is required is the definition of the three problem specific components, which are also required in the standard SSVMs. As will be shown later, only the loss-augmented inference step becomes harder compared to SSVMs due to the inclusion of privileged information.

---

**Algorithm 1** Cutting plane method for solving Eq. (6)

---

1: **Input:** $(\boldsymbol{x}_1, \boldsymbol{x}_1^*, \boldsymbol{y}_1), \ldots, (\boldsymbol{x}_n, \boldsymbol{x}_n^*, \boldsymbol{y}_n), C, \rho, \gamma, \epsilon$
2: $S_i \leftarrow \emptyset$ for all $i = 1, \ldots, n$
3: **repeat**
4:    **for** $i = 1, \ldots, n$ **do**
5:       SET-UP SURROGATE TASK-SPECIFIC LOSS (EQ. (7))
6:       $\bar{\Delta}(\boldsymbol{y}_i, \boldsymbol{y}, \boldsymbol{y}^*) = \frac{1}{\rho}\Delta^*(\boldsymbol{y}_i, \boldsymbol{y}^*) + [\Delta(\boldsymbol{y}_i, \boldsymbol{y}) - \Delta^*(\boldsymbol{y}_i, \boldsymbol{y}^*)]_+$
7:       SET-UP COST FUNCTION (EQ. (12))
8:       $H(\boldsymbol{y}, \boldsymbol{y}^*) = \bar{\Delta}(\boldsymbol{y}_i, \boldsymbol{y}, \boldsymbol{y}^*) - \langle \boldsymbol{w}, \delta\Psi_i(\boldsymbol{y}) \rangle - \langle \boldsymbol{w}^*, \delta\Psi_i^*(\boldsymbol{y}^*) \rangle$
9:       FIND CUTTING PLANE
10:      $(\hat{\boldsymbol{y}}, \hat{\boldsymbol{y}}^*) = \arg\max_{\boldsymbol{y}, \boldsymbol{y}^* \in \mathcal{Y}} H(\boldsymbol{y}, \boldsymbol{y}^*)$
11:      FIND VALUE OF CURRENT SLACK
12:      $\xi_i = \max\{0, \max_{\boldsymbol{y}, \boldsymbol{y}^* \in S_i} H(\boldsymbol{y}, \boldsymbol{y}^*)\}$
13:      **if** $H(\hat{\boldsymbol{y}}, \hat{\boldsymbol{y}}^*) > \xi_i + \epsilon$ **then**
14:         ADD CONSTRAINT TO WORKING SET
15:         $S_i \leftarrow S_i \cup \{(\hat{\boldsymbol{y}}, \hat{\boldsymbol{y}}^*)\}$
16:         $(\boldsymbol{w}, \boldsymbol{w}^*) \leftarrow$ optimize Eq. (6) over $\cup_i S_i$.
17:      **end if**
18:    **end for**
19: **until** no $S_i$ has changed during iteration

---

### 3.2 Joint Feature Map

Our extended structured output regressor, SSVM+, estimates bounding box coordinates within target images by considering all possible bounding boxes. The structured output space is defined as $\mathcal{Y} \equiv \{(\theta, t, l, b, r) \mid \theta \in \{+1, -1\}, (t, l, b, r) \in \mathbb{R}^4\}$, where $\theta$ denotes the presence/absence of an object and $(t, l, b, r)$ correspond to coordinates of the top, left, bottom, and right corners of a bounding box, respectively. To model the relationship between input and output variables, we define a joint feature map, encoding features in $\boldsymbol{x}$ to their bounding boxes defined by $\boldsymbol{y}$. This is modeled as

$$\Psi(\boldsymbol{x}_i, \boldsymbol{y}) = \boldsymbol{x}_i|_{\boldsymbol{y}}, \tag{8}$$

where $\boldsymbol{x}|_{\boldsymbol{y}}$ denotes the region of an image inside a bounding box with coordinates $\boldsymbol{y}$. Identically, for the privileged space, we define another joint feature map, which instead of on visual features, it operates on the space of attributes aided by segmentation information as

$$\Psi^*(\boldsymbol{x}_i^*, \boldsymbol{y}^*) = \boldsymbol{x}_i^*|_{\boldsymbol{y}^*}. \tag{9}$$

The definition of the joint feature map is problem specific, and we follow the method in [1] proposed for object localization. Implementation details about both joint feature maps are described in Section 4.2

### 3.3 Task-Specific Loss

To measure the level of discrepancy between the predicted output $\boldsymbol{y}$ and the true structured label $\boldsymbol{y}_i$, we need to define a loss function that accurately measures such a level of disagreement. In our object localization problem, the following task-specific loss, based on the Pascal VOC overlap ratio [1], is employed in both spaces,

$$\Delta(\boldsymbol{y}_i, \boldsymbol{y}) = \begin{cases} 1 - \frac{\mathrm{area}(\boldsymbol{y}_i \cap \boldsymbol{y})}{\mathrm{area}(\boldsymbol{y}_i \cup \boldsymbol{y})} & \text{if } \boldsymbol{y}_{i\theta} = \boldsymbol{y}_\theta = 1 \\ 1 - (\frac{1}{2}(\boldsymbol{y}_{i\theta}\boldsymbol{y}_\theta + 1)) & \text{otherwise,} \end{cases} \tag{10}$$

where $\boldsymbol{y}_{i\theta} \in \{+1, -1\}$ denotes the presence $(+1)$ or absence $(-1)$ of an object in the $i$-th image. In the case $\boldsymbol{y}_{i\theta} = -1$, $\Psi(\boldsymbol{x}|_{\boldsymbol{y}}) = \boldsymbol{0}$, where $\boldsymbol{0}$ is an all zero vector. The loss is 0 when bounding boxes defined by $\boldsymbol{y}_i$ and $\boldsymbol{y}$ are identical, and equal to 1 when they are disjoint or $\boldsymbol{y}_{i\theta} \neq \boldsymbol{y}_\theta$.

### 3.4 Loss-Augmented Inference

Due to the exponential number of constraints that arise during learning of Eq. (6) and the possibly very large search space $\mathcal{Y}$ dealt with during prediction, we require an efficient inference technique, which may differ in training and testing in the SSVM+ framework.

### 3.4.1 Prediction

The goal is to find the best bounding box given the learned weight vector $\boldsymbol{w}$ and the visual feature $\boldsymbol{x}$. Privileged information is not available at testing time, and inference is performed on visual features only. Therefore, the same maximization problem as in standard SSVMs needs to be solved during prediction, which is given by

$$h(\boldsymbol{x}) = \arg\max_{\boldsymbol{y} \in \mathcal{Y}} \langle \boldsymbol{w}, \Psi(\boldsymbol{x}, \boldsymbol{y}) \rangle. \tag{11}$$

This maximization problem is over the space of bounding box coordinates. However, this problem involves a very large search space and therefore cannot be solved exhaustively. In the object localization task, the Efficient Subwindow Search (ESS) algorithm [2] is employed to solve the optimization problem efficiently.

### 3.4.2 Learning

Compared to the inference problem required during the prediction step shown in Eq. (11), the optimization of our main objective during training involves a more complex inference procedure. We need to perform the following maximization with the surrogate loss and an additional term corresponding to the privileged space during an iterative procedure:

$$
\begin{aligned}
(\hat{\boldsymbol{y}}, \hat{\boldsymbol{y}}^*) &= \arg\max_{\boldsymbol{y}, \boldsymbol{y}^* \in \mathcal{Y}} \bar{\Delta}(\boldsymbol{y}_i, \boldsymbol{y}, \boldsymbol{y}^*) - \langle \boldsymbol{w}, \delta\Psi_i(\boldsymbol{y}) \rangle - \langle \boldsymbol{w}^*, \delta\Psi_i^*(\boldsymbol{y}^*) \rangle \\
&= \arg\max_{\boldsymbol{y}, \boldsymbol{y}^* \in \mathcal{Y}} \bar{\Delta}(\boldsymbol{y}_i, \boldsymbol{y}, \boldsymbol{y}^*) + \langle \boldsymbol{w}, \Psi(\boldsymbol{x}_i, \boldsymbol{y}) \rangle + \langle \boldsymbol{w}^*, \Psi^*(\boldsymbol{x}_i^*, \boldsymbol{y}^*) \rangle.
\end{aligned} \tag{12}
$$

Note that $\langle \boldsymbol{w}, \Psi(\boldsymbol{x}_i, \boldsymbol{y}_i) \rangle$ and $\langle \boldsymbol{w}^*, \Psi^*(\boldsymbol{x}_i^*, \boldsymbol{y}_i) \rangle$ are constants in Eq. (12) and do not affect the optimization. The problem in Eq. (12), called loss-augmented inference, is required during each iteration of the cutting plane method, which is used for learning the functions $h$ and $\phi$ and hence the weight vectors $\boldsymbol{w}$ and $\boldsymbol{w}^*$.

We adopt an alternating approach for the inference, where we first solve for $\boldsymbol{y}^*$ in the privileged space given the fixed solution in the original space $\boldsymbol{y}_c$

$$\arg\max_{\boldsymbol{y}^* \in \mathcal{Y}} \bar{\Delta}(\boldsymbol{y}_i, \boldsymbol{y}_c, \boldsymbol{y}^*) + \langle \boldsymbol{w}^*, \Psi^*(\boldsymbol{x}_i^*, \boldsymbol{y}^*) \rangle \tag{13}$$

and subsequently perform optimization in the original space while fixing $\boldsymbol{y}_c^*$

$$\arg\max_{\boldsymbol{y} \in \mathcal{Y}} \bar{\Delta}(\boldsymbol{y}_i, \boldsymbol{y}, \boldsymbol{y}_c^*) + \langle \boldsymbol{w}, \Psi(\boldsymbol{x}_i, \boldsymbol{y}) \rangle. \tag{14}$$

These two sub-procedures in Eq. (13) and (14) are repeated until convergence, and we obtain the final solutions $\boldsymbol{w}$ and $\boldsymbol{w}^*$. In the object localization task, both problems are solved by ESS [2], a branch-and-bound optimization technique, for which it is essential to derive upper bounds of the above objective functions over a set of rectangles from $\mathcal{Y}$. Here we derive the upper bounds of only the surrogate loss terms in Eq. (7); the derivation for the other terms can be found in [2].

When the solution in the privileged space is fixed, we need to consider the upper bound of only $[\Delta - \Delta^*]_+$ to obtain the upper bound of the surrogate loss. Since $[\Delta - \Delta^*]_+$ is a monotonically increasing function of $\Delta$, its upper bound is derived directly from the upper bound of $\Delta$. Specifically, the upper bound of $\Delta$ is given by

$$\Delta = 1 - \frac{\mathsf{area}(\boldsymbol{y}_i \cap \boldsymbol{y})}{\mathsf{area}(\boldsymbol{y}_i \cup \boldsymbol{y})} \leq 1 - \frac{\min_{\boldsymbol{y} \in \mathcal{Y}} \mathsf{area}(\boldsymbol{y}_i \cap \boldsymbol{y})}{\max_{\boldsymbol{y} \in \mathcal{Y}} \mathsf{area}(\boldsymbol{y}_i \cup \boldsymbol{y})}, \tag{15}$$

and the upper bound of the surrogate loss with a fixed $\Delta^*$ is given by

$$[\Delta - \Delta^*]_+ \leq \left[ 1 - \frac{\min_{\boldsymbol{y} \in \mathcal{Y}} \mathsf{area}(\boldsymbol{y}_i \cap \boldsymbol{y})}{\max_{\boldsymbol{y} \in \mathcal{Y}} \mathsf{area}(\boldsymbol{y}_i \cup \boldsymbol{y})} - \Delta^* \right]_+. \tag{16}$$

When the original space is fixed, the problem is not straightforward since the surrogate loss becomes a V-shaped function with $\rho > 1$. In this case, we need to check outputs of the function at both upper

and lower bounds of $\Delta^*$. The upper bound of $\Delta^*$ is derived identically to that of $\Delta$, and the lower bound of $\Delta^*$ is given by

$$\Delta^* = 1 - \frac{\text{area}(\boldsymbol{y}_i \cap \boldsymbol{y}^*)}{\text{area}(\boldsymbol{y}_i \cup \boldsymbol{y}^*)} \geq 1 - \frac{\max_{\boldsymbol{y}^* \in \mathcal{Y}} \text{area}(\boldsymbol{y}_i \cap \boldsymbol{y}^*)}{\min_{\boldsymbol{y}^* \in \mathcal{Y}} \text{area}(\boldsymbol{y}_i \cup \boldsymbol{y}^*)}. \tag{17}$$

Let $\Delta_u^*$ and $\Delta_l^*$ be the upper and lower bounds of $\Delta^*$, respectively. Then the upper bound of the surrogate loss with a fixed $\Delta$ is given by

$$\frac{1}{\rho}\Delta^* + [\Delta - \Delta^*]_+ \leq \max\left( \frac{1}{\rho}\Delta_u^* + [\Delta - \Delta_u^*]_+, \ \frac{1}{\rho}\Delta_l^* + [\Delta - \Delta_l^*]_+ \right). \tag{18}$$

By identifying the bounds of the surrogate loss as in Eq. (17) and (18), we can optimize the objective function in Eq. (12) through the alternating procedure based on the standard ESS algorithm.

## 4 Experiments

### 4.1 Dataset

Empirical evaluation of our method is performed on the Caltech-UCSD Birds 2011 (CUB-2011) [5] fine-grained categorization dataset. It contains 200 categories of different species of birds. The location of each bird is specified using a bounding box. In addition, a large collection of privileged information is provided in the form of 15 different part annotations, 312 attributes and segmentation masks, manually labeled in each image by human annotators. Each category contains 30 training images and around 30 testing images.

### 4.2 Visual and Privileged Feature Extraction

Our feature descriptor in visual space adopts the bag-of-visual-words model based on Speeded Up Robust Features (SURF) [26], which is almost identical to [2]. The dimensionality of visual feature descriptors is 3,000. We additionally employ attributes and segmentation masks as privileged information. The information about attributes is described by a 312 dimensional vector, whose element corresponds to each attribute and which has a binary value depending on its visibility and relevance. We use segmentation information to inpaint segmentation masks into each image, which results in an image containing the original background pixels with uniform foreground pixels. Subsequently, we extract the 3,000-dimensional feature descriptor based on the same bag-of-visual-words model as in the visual space. The intuition behind this approach is to generate a set of features that provide a guaranteed strong response in the foreground region. This response is to be stronger than in the original space, hence allowing for easier localization in the privileged space. For each sub-window, we create a histogram based on the presence of attributes and the frequency of the privileged codewords corresponding to the augmented visual space.

### 4.3 Evaluation

To evaluate our SSVM+ algorithm, we compare it against the original SSVM localization method by Blaschko and Lampert [1] in several training scenarios. In all experiments we tune the hyper-parameters $C$, $\lambda$ and $\rho$ on a $4 \times 4 \times 4$ space spanning values $[2^{-8}, ..., 2^5]$. For SSVM, one dimension of the search space corresponding to the parameter $C$ is searched.

We first investigate the influence of small training sample sizes on localization performance. For this setting, we loosely adopt the experimental setup of [27]. For training, we focus on 14 bird categories corresponding to 2 major bird groups. We train four different models, each trained on a distinctive number of training images, namely $n_c = \{1, 5, 10, 20\}$ images per class, resulting in $n = \{14, 70, 140, 280\}$ training images, respectively. Additionally, we train a model on $n = 1000$ images, corresponding to 100 bird classes, each with 10 training images. As a validation set, 500 training images chosen at random from categories other than the ones used for training are used. For testing, we use all testing images of the entire CUB-2011 dataset. Table 1 presents results of this experiment. In all cases, our method outperforms the SSVM method in both average overlap as well as average detection (PASCAL VOC overlap ratio $> 50\%$). This implies that for

Table 1: Comparison between our SSVM+ and the standard SSVM [1] by varying the number of classes and training images.

| | (A) OVERLAP | | | | | (B) DETECTION | | | | |
|---|---|---|---|---|---|---|---|---|---|---|
| # training images | 14 | 70 | 140 | 280 | 1000 | 14 | 70 | 140 | 280 | 1000 |
| SSVM [1] | 38.2 | 43.8 | 42.3 | 44.9 | 48.1 | 25.9 | 37.3 | 34.3 | 39.8 | 46.2 |
| SSVM+ | **41.3** | **45.7** | **45.8** | **46.9** | **49.0** | **32.6** | **42.4** | **41.5** | **43.3** | **48.1** |
| DIFF. | **+3.1** | **+1.9** | **+3.5** | **+2.0** | **+0.9** | **+6.7** | **+5.1** | **+7.2** | **+3.5** | **+1.9** |

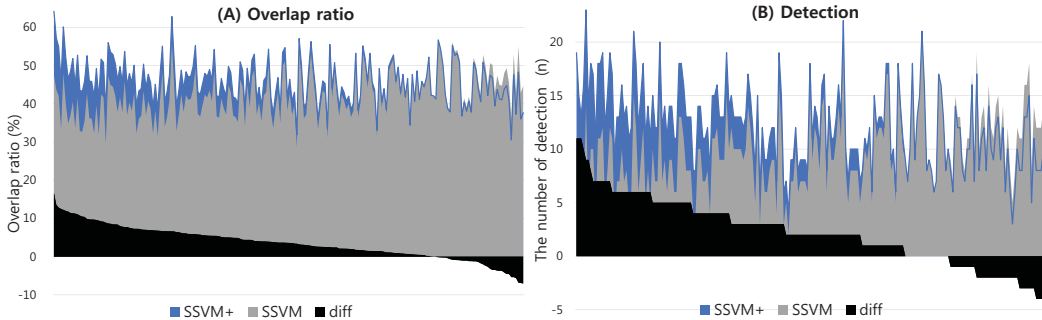

Figure 2: Comparison results of average overlap (**A**) and detection results (**B**) between our structured learning with privileged information (SSVM+) and the standard structured learning (SSVM) on 100 classes of the CUB-2011 dataset. The bird classes aligned in $x$-axis are sorted by the differences of two methods shown in black area in a non-increasing order.

the same number of training examples, our method consistently converges to a model with better generalization performance than SSVM. A previously observed trend [4, 23] of decreasing benefit of privileged information with increasing training set sizes is also apparent here.

To evaluate the benefit of SSVM+ in more depth, we illustrate average overlap and detection performance on all the 100 classes in Figure 2, where 10 images per class are used for training with 14 classes ($n = 140$). In most of bird classes, SSVM+ shows relatively better performance in both overlap ratio and detection rate. Note that each class typically has 30 testing images but some classes have as little as 18 images. Average overlap ratio is $45.8\%$ and average detection is 12.1 ($41.5\%$).

## 5 Discussion

We presented a structured prediction algorithm for object localization based on SSVM with privileged information. Our algorithm is the first method for incorporating privileged information within a structured prediction framework. Our method allows the use of various types of additional information during training to improve generalization performance at testing time. We applied our proposed method to an object localization problem, which is solved by a novel structural SVM formulation using privileged information. We employed an alternating loss-augmented inference procedure to handle the term in the objective function corresponding to privileged information. We applied the proposed algorithm to the Caltech-UCSD Birds 200-2011 dataset and obtained encouraging results, suggesting the potential benefit of exploiting additional information that is available during training only. Unfortunately, the benefit of privileged information tends to reduce as the number of training examples increases; our SSVM+ framework would be particularly useful when there exist only a few training data or annotation cost is very high.

### Acknowledgement

This work was supported partly by ICT R&D program of MSIP/IITP [14-824-09-006; 14-824-09-014] and IT R&D Program of MKE/KEIT (10040246).

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
