[Reviews · NeurIPS 2014]

Submitted by Assigned_Reviewer_8

- The method is a fairly direct generalization of the framework of learning with privileged information to the structured output case. The method is effective for the object localization task and results in good improvements in localization accuracy.
- Does this formulation of SSVM+ reduce to SVM+ for binary classification with the right choice of loss function and feature vector, just as SSVM reduce to SVM in the classical case? It looks like the authors' formulation of SSVM+ contains separate slack variables \xi_i for each example x_i and there are extra degrees of freedom.
- The loss functions \Delta and \Delta^* seems to be the same in the object localization task, but it is not stated clearly in section 3.3.
- How effective is the alternating inference procedure between y and y^*? How many alternating iterations are required? When the parameter vectors w and w^* are far from the optimal solution, could this alternating inference procedure get stuck in bad local minima?
- I am interested in knowing the strength of the privileged information encoded in the object localization task. If we train a regular SSVM with the privileged information and also use the privileged information during testing, how well will the algorithm do? This can give us some understanding on how informative the privileged features are when they're available during test time, vs only during training time under the LUPI framework.
- Overall I think the method is a simple extension of the learning under privileged information framework, but it works well. It can be improved if the authors can give more explanations of their formulation and how it links to the case of binary SVM+, or give some theoretical justification like those under the LUPI framework.
Summary: The authors proposed an extension of learning with privileged information (i.e., extra features that are not available during test time) framework from classification to structured prediction. The extension is fairly simple, but is effective for the object localization task in improving localization accuracy. The authors also give algorithm for performing inference during training and a cutting plane algorithm for solving the optimization problem.

Submitted by Assigned_Reviewer_15

The paper presents a method for structured output prediction using privileged information, and applied the method to object localization task. The benefit of the method is that in learning time richer training data (more expensive measurements, for example) can be used than what is available during testing time, while still improving the generalisation accuracy on future data. The authors demonstrate the performance gains in empirical experiments against Structured SVM without privileged data.

Quality: The approach of the paper is solid in my view, and the argumentation is convincing. However, there is not much theoretical discussion, e.g. improvement of convergence rates in the structured output case.

Clarity: The paper is written in reasonably clear way, easy to read even for a person not working on image data.

Originality: The generalisation of the LUPI framework to structured outputs is new to my knowledge. As such it falls rather nicely to the Vapnik's original framework for binary classification without too much trouble.

Significance: The SSVM+ method is useful in the present application of object localisation, and may find use in other application fields of structured output. However, the current paper is not exploring the wider applications.
Summary: The paper show how Vapnik's LUPI framework can be generalised to structured output, in this case object localisation. Empirical performance gains are observed.

Submitted by Assigned_Reviewer_22

The work introduces a learning framework that combines structured prediction
[Tsochantaridis et al. Large margin methods for structured and interdependent
output variables.] and SVM learning using privileged information (LUPI) [4]. A
novel training objective and a method to do loss-augmented inference is
presented. The paper closes with positive results on the Caltech-UCSD Birds
200-2011 dataset.

Quality: The work clearly motivates the newly introduced learning framework and
combines the two used frameworks of structured prediction and LUPI. The
experiments seem to support the claims of improved generalization compared to
the training without any privileged information. Futhermore, the paper shows a
rather generic way to make use of additional training segmentation masks for
object detection.

Clarity: The paper seems well polished and the claims can be well followed.

Originality: A combination of the two frameworks seems to be novel.

Significance: Learning with few training data, with added context information
is important for structured prediction.

Strength:
+ Clear presentation and interesting problem.
+ Successful application to a object detection task.
Summary: The work approaches the interesting question on structured learning with additional information. The proposed framework is well motivated and is clearly stated.
Author Feedback
Author rebuttal: We would sincerely like to thank all reviewers for their positive and constructive comments. Below, we separately address reviewers’ main points.

-- Assigned_Reviewer_15 --

1. “Convergence rate of SSVM+”
Analysis of the convergence rate of SSVM+ to the best possible solution in the admissible set of solutions is not trivial. Even in the case of SVM+, a risk bound has not yet been derived. Vapnik derived bounds (Propositions 1 and 2) for the unregularized learning with privileged information based on zero-one loss (Eq. (10) and (11)) in [4]. A related result, but with a more general bound, was found in [23] for the case of empirical risk minimization (ERM) with privileged information. Our SSVM+ method is a regularized form of ERM, and hence their analysis is relevant when regularization is not considered. We are currently working on the analysis of both SVM+ and SSVM+ in this regard.

-- Assigned_Reviewer_8 --

1. “Relationship between SSVM+ and SVM+”
Under the conditions in Eq. (28) in [Ref1] and \rho = 1, our SSVM+ formulation can be reduced to a max-margin binary classifier using privileged information, which is similar but not exactly the same as the original SVM+. In contrast to SVM+, the reduced version of SSVM+ includes corrected slack variables for each training example; the slack \xi_i in each constraint is replaced by "\xi_i + y < w^*, x^* > " which means that prediction in the privileged space controls margin in the original space, as the original SVM+ does.

2. “Alternating inference”
In our experiments, the alternating optimization (Eq. (13) and Eq. (14)) converged to a solution very quickly, in only a few iterations in most cases (more than 95% of the entire trials). Empirically, we have observed that the method achieves a good solution after just one iteration, improving upon the solution slightly in subsequent iterations. Each iteration of the alternating optimization method corresponds to the ESS [2] procedure, with our problem-specific upper bound, and hence a global solution w.r.t Eq. (13) or Eq. (14) is found at every step [2]. In our experiments, we have not observed the algorithm getting stuck at bad local minima, even though the initial values of w and w^* can be arbitrary, depending on the training set.

3. “How useful is Privileged Information?”
We evaluated SSVM with original and privileged features, both available during training and prediction. Its average overlap ratio was better than that of our SSVM+ by about 10%, which shows the strength of the privileged information for object localization.

4. “Delta and Delta^*”
Both losses are the PASCAL VOC overlap ratio; we will make this clear in the revised version if our paper is accepted.

[References]:
[Ref1] Thorsten Joachims, Thomas Finley, Chun-Nam John Yu, Cutting-plane training of structural SVMs, Machine Learning, Volume 77, Issue 1, pp 27-59